# Revisiting Depth-guided Methods for Monocular 3D Object Detection by Hierarchical Balanced Depth

**Yi-Rong Chen**
National Taiwan University, Taiwan
`andy94077s@cmlab.csie.ntu.edu.tw`

**Ching-Yu Tseng**
National Taiwan University, Taiwan
`cytseng@cmlab.csie.ntu.edu.tw`

**Yi-Syuan Liou**
National Taiwan University, Taiwan
`yisyuanliou@cmlab.csie.ntu.edu.tw`

**Tsung-Han Wu**
National Taiwan University, Taiwan
`tsunghan@cmlab.csie.ntu.edu.tw`

**Winston H. Hsu**
National Taiwan University, Taiwan
`whsu@ntu.edu.tw`

**Abstract:** Monocular 3D object detection has seen significant advancements with the incorporation of depth information. However, there remains a considerable performance gap compared to LiDAR-based methods, largely due to inaccurate depth estimation. We argue that this issue stems from the commonly used pixel-wise depth map loss, which inherently creates the imbalance of loss weighting between near and distant objects. To address these challenges, we propose MonoHBD (**Mono**cular **H**ierarchical **B**alanced **D**epth), a comprehensive solution with the hierarchical mechanism. We introduce the Hierarchical Depth Map (HDM) structure that incorporates depth bins and depth offsets to enhance the localization accuracy for objects. Leveraging RoIAlign, our Balanced Depth Extractor (BDE) module captures both scene-level depth relationships and object-specific depth characteristics while considering the geometry properties through the inclusion of camera calibration parameters. Furthermore, we propose a novel depth map loss that regularizes object-level depth features to mitigate imbalanced loss propagation. Our model reaches state-of-the-art results on the KITTI 3D object detection benchmark while supporting real-time detection. Excessive ablation studies are also conducted to prove the efficacy of our proposed modules.

**Keywords:** monocular 3D object detection, autonomous driving

## 1 Introduction

Accurately detecting and locating objects in 3D space at an affordable price is crucial for the autonomous vehicle and robotic industries. Despite the emergence of various monocular-based approaches [1, 2, 3], which rely on a single image for 3D detection, their performance still lags behind conventional LiDAR methods [4, 5, 6] due to the inaccurate depth information. To bridge the performance gap, modern depth-guided solutions [7, 8, 9, 10, 11] have introduced modules to predict depth distributions with auxiliary depth supervision. In particular, these approaches commonly employ linear-increasing discretization (LID) [12] along with focal loss [13] to supervise the depth estimation. However, these models still suffer from poor depth estimation as the depth increases [10], resulting in inferior performance.

In this paper, we argue that the limitations of depth-guided methods stem from the inherent structure of conventional pixel-wise depth maps. We identify two common weaknesses:

7th Conference on Robot Learning (CoRL 2023), Atlanta, USA.

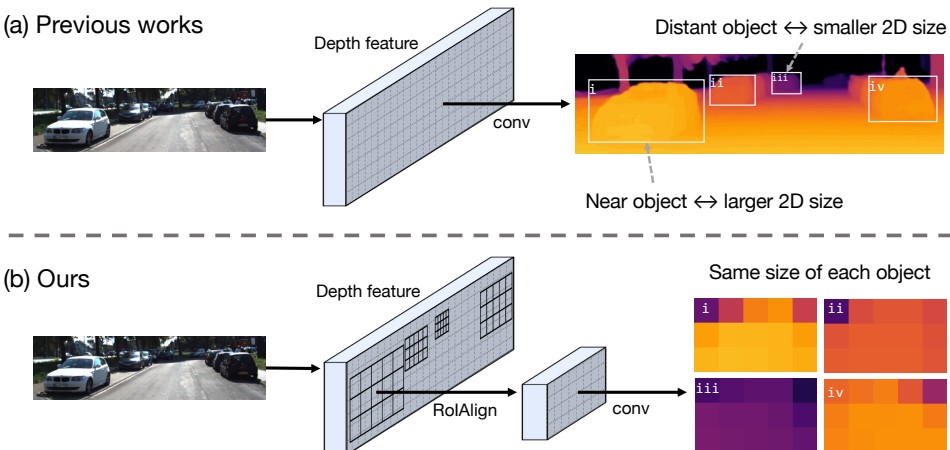

Figure 1: Comparison of depth map structures between various depth-guided monocular 3D object detection methods. (a) Previous works measure the pixel-wise losses directly on the depth map, creating an inherent weight imbalance problem due to the significant size difference between near and distant objects. (b) Our method extracts the object-wise feature from the depth feature using RoIAlign [14], effectively addressing the issue of complexity between depth and objects' 2D size.

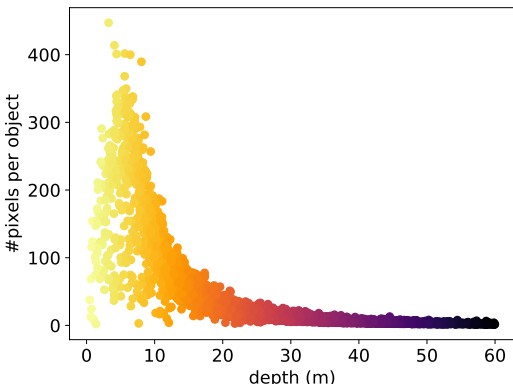

Figure 2: The distribution of the objects' depth in the KITTI dataset and their 2D object areas. Near objects have larger 2D sizes than distant ones, leading to inherent imbalanced loss weights of each object for common pixel-wise losses. To address this problem, we propose BDE (Section 3.4) to create balanced depth maps.

(1) **Imbalanced loss weighting among objects with different depths.** Current depth map losses, such as cross-entropy loss and focal loss, are computed at the pixel level, creating an inherent imbalance between distant and near objects (Figure 1(a)). This is because distant objects appear smaller on the depth map compared to near objects (Figure 2), resulting in distant objects receiving less weight in pixel-wise losses.

(2) **Ambiguous depth distribution of depth maps.** The linear-increasing discretization (LID) [12] method partitions depths into depth bins, promoting the learning of depth distributions rather than a single depth value. However, this method suffers from a wider bin range as the depth increases, which introduces ambiguous localization of distant objects.

To address these problems, we propose a comprehensive solution – MonoHBD. We introduce the Hierarchical Balanced Depth mechanism (Figure 1(b)) specifically for 3D object detection. To be exact, we present a novel Hierarchical Depth Map (HDM) structure (Section 3.3) that extends the conventional depth discretization method by incorporating the prediction of offsets for each depth bin. Additionally, we propose the Regression Focal Loss (RFL) to simultaneously regularize depth

bins and offsets. With HDM and RFL, our model offers a more precise depth representation of depth maps and boosts the overall performance (Section 4.3).

Furthermore, we propose a novel depth predictor module Balanced Depth Extractor (BDE) (Section 3.4) that untangles the complexity between depth and object size within the depth map. Leveraging RoIAlign [14], we extract valuable object depth features from the depth predictor and estimate their depth distribution. Notably, the extracted depth maps maintain a consistent size regardless of the 2D bounding box size of objects. This fundamentally resolves the challenge of imbalanced pixel counts across objects with varying depths. By employing BDE, we transform the regression problem from the pixel level to the object level, leading to a more suitable approach for 3D object detection tasks.

In summary, our contributions are as follows:

1. We are the first work that identifies the issue of imbalanced pixel counts for objects at different depths, which is problematic for current depth losses in 3D object detection. To address this, we propose a Balanced Depth Extractor (BDE) module that shifts the focus from pixel-level regression to object-level regression.

2. We introduce a Hierarchical Depth Map (HDM) structure that utilizes a coarse-to-fine approach, enabling more accurate depth estimation for the depth map. The Regression Focal Loss (RFL) is proposed to regress the HDM efficiently and effectively.

3. Our approach achieves state-of-the-art results on the KITTI dataset and supports real-time detection. Also, ablation studies validate the efficacy of all the proposed modules.

## 2   Related Work

### 2.1   Image-only Monocular 3D Object Detection

In recent years, there has been significant research interest in monocular 3D object detection, driven by its cost-effectiveness and simplicity in setup. These methods leverage geometric constraints to predict 3D bounding boxes from single images. For instance, Deep3DBox [15] incorporates geometric priors and a multi-bin loss to tackle angle prediction. Deep MANTA [16] utilizes 3D CAD models to aid in 3D bounding box prediction. M3D-RPN [1] introduces a depth-aware convolution module and exploits geometric constraints between 2D and 3D bounding boxes. MonoPair [2] handles occluded objects by considering pairwise relationships between samples. FCOS3D [17] extends the anchor-free FCOS [18] to 3D object detection. OFT [19] employs orthographic feature transform to project image features into the LiDAR coordinate space. GUPNet [3] incorporates the Geometry Uncertainty Projection module to mitigate the error amplification effect. However, the performance of these methods is limited due to the lack of depth information from a single monocular image.

### 2.2   Depth-guided Monocular 3D Object Detection

Depth-guided monocular 3D object detection is an alternative approach that leverages depth information to enhance detection performance. Pseudo-LiDAR based methods [20, 21] convert image features to a pseudo-LiDAR representation using a pre-trained depth estimation model, such as DORN [22]. CaDDN [10] introduces a categorical depth distribution with auxiliary supervision. DD3D [11] learns depth-aware features by pre-training on large-scale datasets, including KITTI-Depth [23] and DDAD15M [11]. MonoDTR [8] injects depth embeddings generated by the Depth-Aware Feature Enhancement module into a transformer network. MonoDETR [9] supervises pixel-wise depth maps with objects' center depth and employs a depth-aware transformer for 3D bounding box prediction. MonoCon [7] learns auxiliary projected 2D signals to enhance 3D detection tasks. While these methods learn depth distribution either implicitly or through depth supervision, the predicted pixel-wise depth maps introduce inherent loss weighting imbalance for near and distant objects, leading to sub-optimal loss propagation for 3D object detection tasks.

# 3 Method

## 3.1 Problem Definition

Given an RGB image $I \in \mathbb{R}^{3 \times H_I \times W_I}$ and its corresponding calibration matrix $K \in \mathbb{R}^{3 \times 4}$, our model aims to predict a set of 3D bounding boxes $\hat{\mathbf{B}} = \{\hat{B}_1, \hat{B}_2, \ldots, \hat{B}_N\}$, where each $\hat{B}_i$ represents a bounding box with the following properties: $c_i$ denotes the object class, $(x_i, y_i, z_i)$ represents the 3D center position in the camera coordinate system, $(h_i, w_i, \ell_i)$ corresponds to the height, width, and length of the bounding box, and $\theta_i$ represents the yaw angle. Note that in monocular 3D object detection, the roll and pitch angles of the bounding box are assumed to be zero.

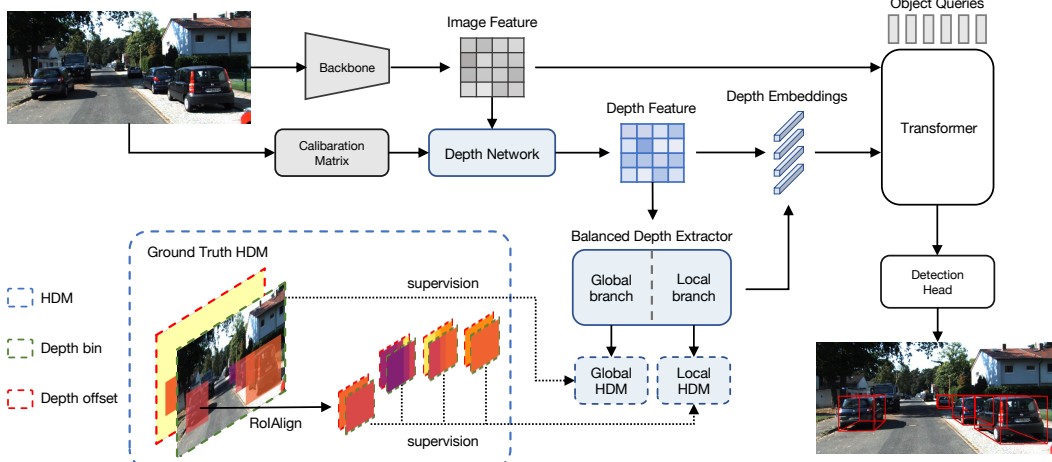

Figure 3: The overall architecture of MonoHBD. The input image is fed to the backbone to extract image features. The depth features are then produced by the depth network with image features and the calibration matrix. Leveraging depth supervision, the Hierarchical Depth Map (HDM) (Section 3.3) for the whole image (Global HDM) and each object (Local HDM) are extracted by the Balanced Depth Extractor (BDE) (Section 3.4). Each HDM is supervised by our Regression Focal Loss (RFL) (Section 3.5). After fusing depth features and the Hierarchical Depth Map to produce the depth embedding, the transformer joins image features and depth embeddings. Lastly, the detection head predicts the 3D bounding boxes (Section 3.5).

## 3.2 Overview and Architecture

Figure 3 is the architecture of our MonoHBD, which is built upon the common depth-guided framework [8, 9, 24]. It consists of the feature backbone, the depth network, the Balanced Depth Extractor (BDE), the depth-aware transformer (DTR) [9], and the detection head. We feed the RGB image into the image backbone (Section 3.2) to get the image features, and then the depth network (Section 3.2) produces the depth features from the image feature and the calibration matrix. The Balanced Depth Extractor (BDE) module (Section 3.4) predicts the depth map hierarchically by the Hierarchical Depth Map (HDM) structure (Section 3.3). After that, a set of learnable object queries are initialized and the image and depth embeddings are injected into the DTR module to perform the cross-attention mechanism. Finally, the detection head (Section 3.5) predicts 3D bounding boxes from the output hidden features of DTR. We provide details of DTR in the supplementary material.

**Feature Backbone.** Given an RGB image $I \in \mathbb{R}^{3 \times H_I \times W_I}$ and the backbone $f_B(\cdot; \theta_B)$, we generate the image feature $F_I \in \mathbb{R}^{C \times H_F \times W_F}$ by $F_I = f_B(I; \theta_B)$. In practice, we adopt the commonly used DLA-34 [25] backbone and set $C = 256, H_F = H_I/16$ and $W_F = W_I/16$ to ensure a fair comparison among previous works.

**Depth Network.** We adopt the depth head from [9] as the prototype of our depth network. However, the original depth head does not consider the camera calibration parameters, leading to incorrect depth distribution prediction. This issue becomes more significant when the input image is scaled

or shifted [26], which is common during training and testing. Therefore, it is crucial to provide the correct calibration parameters to the model to establish a better spatial understanding.

Drawing inspiration from [26], we first flatten the calibration matrix $K$ and then pass it through a multi-layer perceptron (MLP) layer. The transformed feature is then concatenated with the image feature $F_I$ to generate the depth feature $F_D$. Formally, $F_D$ is obtained by:

$$F_D = f_D(F_I \oplus MLP(flatten(K)); \theta_D), \tag{1}$$

where $F_D \in \mathbb{R}^{C \times H_F \times H_F}$, $\oplus$ denotes the concatenation operation, and $f_D(\cdot; \theta_D)$ represents the depth network with parameters $\theta_D$.

### 3.3 Hierarchical Depth Map

We propose the Hierarchical Depth Map (HDM) structure for more accurate depth estimation. Previous works employ the linear-increasing discretization (LID) [12] method to divide depths into classes. LID is defined as:

$$\text{LID}(d) = \left\lfloor -\frac{1}{2} + \frac{1}{2}\sqrt{1 + \frac{4N_D(N_D+1)}{d_{max} - d_{min}} \cdot (d - d_{min})} \right\rfloor \tag{2}$$

$$\text{LID}^{-1}(i) = d_i = d_{min} + \frac{d_{max} - d_{min}}{N_D(N_D+1)} \cdot i(i+1), \tag{3}$$

where $d_i$ is the depth value for the $i^{th}$ depth bin, $[d_{min}, d_{max}]$ is the desired depth range for discretization, and $N_D$ is the number of depth bins.

Equation 3 shows that distant bins are wider than nearer bins, which means distant objects may suffer from the loss of depth information after the discretization. To address this problem, we introduce the Hierarchical Depth Map (HDM), which consists of a depth bin class map $D_b$ and a depth offset map $D_\delta$. Given a depth map $D \in \mathbb{R}^{H_F \times W_F}$, we construct the Hierarchical Depth Map $\mathcal{H}$ as follows:

$$\begin{aligned}
&\mathcal{H} = (\mathcal{D}, \delta), \\
&\text{where } \mathcal{D}, \delta \in \mathbb{R}^{H_F \times W_F}, \\
&\mathcal{D}(i,j) = \text{LID}(\mathcal{D}(i,j)), \ \delta(i,j) = D(i,j) - \text{LID}^{-1}(\mathcal{D}(i,j))
\end{aligned} \tag{4}$$

We discretize $D$ into the depth bin class map $\mathcal{D}$ using the LID method, and then compute the depth offset map $\delta$ by subtracting the corresponding depth value of the $\mathcal{D}(i,j)$ bin from $D(i,j)$. Compared to LID, HDM adopts a coarse-to-fine approach and preserves the depth details of objects, resulting in improved accuracy and more precise depth estimation for various objects and depth ranges.

### 3.4 Balanced Depth Extractor

Our proposed Balanced Depth Extractor (BDE) predicts the depth distribution in two branches: the global depth branch $f^G$ and the local object-wise depth branch $f^L$. $f^G$ predicts the pixel-wise depth distribution, which regresses the depth of the whole scene and retains the geometry relationships among objects. In contrast, $f^L$ focuses on the depth of foreground objects and eliminates the object size difference regardless of the object depth, leading to a more balanced loss weighting for each object. Note that $f^G$ and $f^L$ are fed with the same depth feature, therefore they are shared weights to maintain the prediction consistency. Also, $f^L$ is only activated during training for guiding depth loss for each object.

We simply feed the depth feature $F_D$ into $f^G$ to generate the HDM $\hat{\mathcal{H}}^G = (\hat{\mathcal{D}}^G, \hat{\delta}^G)$, where $\hat{\mathcal{D}}^G, \hat{\delta}^G \in \mathbb{R}^{(N_D+1) \times H_F \times W_F}$. For $f^L$, we first extract the depth feature for each object using RoIAlign [14] and its ground truth 2D bounding boxes. After that, we input the extracted feature and produce the HDM $\hat{\mathcal{H}}_i^L = (\hat{\mathcal{D}}_i^L, \hat{\delta}_i^L)$ for each object, where $\hat{\mathcal{D}}_i^L, \hat{\delta}_i^L \in \mathbb{R}^{(N_D+1) \times H_G \times W_G}$, and $H_G, W_G$ is the extracted grid size. Importantly, we model the depth offsets independently for each depth bin because the offset distribution among different depth bins may be different.

Finally, we construct the final depth map $\hat{D}^{G*} \in \mathbb{R}^{H_F \times W_F}, \hat{D}_i^{L*} \in \mathbb{R}^{H_G \times W_G}$ from $\hat{\mathcal{H}}^G$ and $\hat{\mathcal{H}}_i^L$ respectively. We employ max-pooling along channel $N_D$ to extract the depth bin and subsequently revert it back to the depth value utilizing Equation 3. Lastly, we add the predicted offsets corresponding to the depth bins to the reverted depth value, resulting in the final depth map.

### 3.5 3D Detection Head and Loss Function

We implement the detection head for 3D object detection based on the approach described in [27]. Following [27], we apply the focal loss [13] for the classification loss $L_{cls}$ to balance the loss weights across different object classes. We utilize the L1 loss for the bounding box regression loss $L_{reg}$.

**Depth Map Classification Loss.** The classification loss of the depth bin distribution is defined as:

$$L_G^{cls} = \frac{1}{H_F W_F} \sum_{i=1}^{H_F} \sum_{j=1}^{W_F} FL(\hat{\mathcal{D}}^G(i,j), \mathcal{D}(i,j)), \tag{5}$$

$$L_L^{cls} = \frac{1}{N_B} \sum_{i=1}^{N_B} \psi_{cls}(\hat{\mathcal{D}}_i^L, \mathcal{D}_i^{RoI}), \tag{6}$$

$$\text{where } \psi_{cls}(\hat{\mathcal{D}}, \mathcal{D}) = \frac{1}{H_G W_G} \sum_{i=1}^{H_G} \sum_{j=1}^{W_G} FL(\hat{\mathcal{D}}(i,j), \mathcal{D}(i,j)), \tag{7}$$

$FL$ is the focal loss [13], $\mathcal{D} \in \mathbb{R}^{H_F \times W_F}$ denotes the ground truth depth bin map, $\mathcal{D}_i^{RoI} \in \mathbb{R}^{H_G \times W_G}$ denotes the ground truth local object-wise depth bin map extracted with the ground truth bounding box $\mathcal{B}_i^{2d}$, and $\hat{\mathcal{D}}^G, \hat{\mathcal{D}}_i^L$ are the depth bin maps predicted by BDE.

**Depth Map Regression Loss.** Extending from the focal loss, we propose the Regression Focal Loss (RFL) for the depth map regression:

$$RFL(p_t, v_t, v) = \alpha_t (1 - p_t)^\gamma \|v_t - v\|, \tag{8}$$

where $\alpha_t$ is the weighting factor, $p_t \in [0, 1]$ is the estimated probability from the model, $\gamma$ is the focusing parameter [13], $v_t \in \mathbb{R}$ is the estimated value from the model, and $v \in \mathbb{R}$ is the ground truth value. The Regression Focal Loss is particularly suited to our task as it concurrently regularizes the depth bins and their offsets. Additionally, since objects closer to the camera tend to have smaller depth offsets due to the narrower depth bins, we introduce the $\alpha_t (1 - p_t)^\gamma$ weighting factor to penalize the imbalanced distribution of depth offsets within each bin.

Therefore, we define the depth map regression loss as:

$$L_G^{reg} = \frac{1}{H_F W_F} \sum_{i=1}^{H_F} \sum_{j=1}^{W_F} RFL(\hat{\mathcal{D}}^G(i,j), \hat{D}^{G*}(i,j), D(i,j)), \tag{9}$$

$$L_L^{reg} = \frac{1}{N_B} \sum_{i=1}^{N_B} \psi_{reg}(\hat{\mathcal{D}}_i^L, \hat{D}_i^{L*}, D_i^{RoI}), \tag{10}$$

$$\text{where } \psi_{reg}(\hat{\mathcal{D}}, \hat{D}^*, D) = \frac{1}{H_G W_G} \sum_{i=1}^{H_G} \sum_{j=1}^{W_G} RFL(\hat{\mathcal{D}}(i,j), \hat{D}^*(i,j), D(i,j)), \tag{11}$$

$\hat{\mathcal{D}}^G, \hat{\mathcal{D}}_i^L$ denote the depth bin maps, and $\hat{D}^{G*}, \hat{D}_i^{L*}$ denote the final depth map (Section 3.4). $D \in \mathbb{R}^{H_F \times W_F}$ is the ground truth depth map, and $D_i^{RoI} \in \mathbb{R}^{H_G \times W_G}$ is the ground truth local object-wise depth map extracted with the ground truth bounding box $\mathcal{B}_i^{2d}$.

Finally, the total loss is computed as:

$$L = \lambda_{cls} L_{cls} + \lambda_{reg} L_{reg} + \lambda_G^{cls} L_G^{cls} + \lambda_L^{cls} L_L^{cls} + \lambda_G^{reg} L_G^{reg} + \lambda_L^{reg} L_L^{reg}, \tag{12}$$

where $\lambda_{cls}, \lambda_{reg}, \lambda_G^{cls}, \lambda_L^{cls}, \lambda_G^{reg}, \lambda_L^{reg}$ are loss weighting hyper-parameters.

Table 1: The detection performance of the Car category on the KITTI test set. The best results are highlighted in red, while the second best ones are highlighted in blue. Note that DD3D [11] is trained with an extra dataset (DDAD15M), which is denoted in *italics*.

| Method | Reference | Time (ms) | AP 3D Car@IoU=0.7 | | | AP BEV Car@IoU=0.7 | | |
|---|---|---|---|---|---|---|---|---|
| | | | easy | mod | hard | easy | mod | hard |
| M3D-RPN [1] | ICCV19 | 160 | 14.76 | 9.71 | 7.42 | 21.02 | 13.67 | 10.23 |
| MonoPair [2] | CVPR20 | 60 | 13.04 | 9.99 | 8.65 | 19.28 | 14.83 | 12.89 |
| PatchNet [30] | ECCV20 | 400 | 15.68 | 11.12 | 10.17 | 22.97 | 16.86 | 14.97 |
| MonoDLE [31] | CVPR21 | 40 | 17.23 | 12.26 | 10.29 | 24.79 | 18.89 | 16.00 |
| Kinematic3D [32] | ECCV20 | 120 | 19.07 | 12.72 | 9.17 | 26.69 | 17.52 | 13.10 |
| CaDDN [10] | CVPR21 | 630 | 19.17 | 13.41 | 11.46 | 27.94 | 18.91 | 17.19 |
| MonoFlex [33] | CVPR21 | 35 | 19.94 | 13.89 | 12.07 | 28.23 | 19.75 | 16.89 |
| GUPNet [3] | ICCV21 | 34 | 22.26 | 15.02 | 13.12 | 30.29 | 21.19 | 18.20 |
| MonoGround [34] | CVPR22 | 30 | 21.37 | 14.36 | 12.62 | 30.07 | 20.47 | 17.74 |
| MonoDTR [8] | CVPR22 | 37 | 21.99 | 15.39 | 12.73 | 28.59 | 20.38 | 17.14 |
| MonoCon [7] | AAAI22 | 26 | **22.50** | **16.46** | **13.95** | **31.12** | **22.10** | **19.00** |
| *DD3D [11]* | *ICCV21* | *-* | *23.19* | *16.87* | *14.36* | *32.35* | *23.41* | *20.42* |
| MonoHBD (Ours) | | 41 | **24.06** | **16.09** | **13.60** | **32.35** | **22.46** | **18.77** |

## 4 Experiment

### 4.1 Setup

**Dataset.** We conduct experiments to evaluate the performance of our model on the widely used KITTI 3D object detection dataset [23]. The dataset consists of a total of 7,481 training images and 7,519 testing images. To ensure fair evaluation, we follow [28] and divide the training images into a training set and a validation set, containing 3,712 and 3,769 samples, respectively. Our ablation studies are conducted using this split.

**Evaluation Metrics.** To evaluate the model performance, we follow the official evaluation settings of the KITTI benchmark. We measure the average precision (AP) of the 3D bounding boxes and the bird's-eye view. Instead of using the 11 recall positions ($AP_{11}$), we utilized the 40 recall positions ($AP_{40}$) to avoid potential bias [29]. The KITTI benchmark defines object difficulties into three levels: "Easy", "Moderate", and "Hard", which are determined based on the height of the 2D bounding boxes, the occlusion level, and the truncation. For cars, we require an Intersection over Union (IoU) score greater than 0.7, while for pedestrians and cyclists, the threshold is set at 0.5.

**Implementation Details.** Our model is trained on a single NVIDIA 3090 GPU with a batch size of 16 for a total of 140 epochs. To generate ground truth depth maps, we empirically utilize the Foreground Depth Map [9]. This choice has been shown to improve model performance. During inference, we omit the local object-wise depth branch in BDE since ground truth bounding boxes are unavailable. Further details are included in the supplementary material.

### 4.2 Main Results

**Performance of the Car category.** Table 1 shows our MonoHBD performance of the Car category compared to other state-of-the-art monocular 3D object detection methods. Our method exhibits state-of-the-art performance on the AP BEV task and delivers competitive results on the AP 3D task. Specifically, our approach outperforms the second-best method MonoCon [7] with an improvement of **1.23 (+4.0%)** and **0.36 (+1.6%)** at the "easy" and "moderate" levels respectively on the AP BEV metric. For the AP 3D metric, we achieve a **1.56 (+6.9%)** improvement at the "easy" level. Significantly, our method offers real-time detection capabilities at 24.4 Frames Per Second (FPS), which demonstrates the efficiency and practical applicability of our approach.

### 4.3 Ablation Study

**Efficacy of each proposed module.** We conduct ablation studies on the KITTI validation set to verify the efficacy of each proposed module in our model. The results are presented in Table 2.

Table 2: Efficacy of each proposed module. $K$ denotes the calibration parameters. $\mathcal{H}$ denotes the HDM (Section 3.3).

| $K$ | $\mathcal{H}$ | RFL | BDE | AP 3D Car@IoU=0.7 | | |
|---|---|---|---|---|---|---|
| | | | | easy | mod | hard |
| | | | | 16.61 | 13.26 | 11.49 |
| ✓ | | | | 21.61 | 16.06 | 13.33 |
| ✓ | ✓ | | | 22.08 | 16.24 | 13.31 |
| ✓ | ✓ | ✓ | | 23.05 | 16.43 | 13.36 |
| ✓ | ✓ | ✓ | ✓ | **23.34** | **16.78** | **13.70** |

Table 3: Comparison of the performance at different depth ranges on the KITTI validation set.

| Method | AP 3D Car@IoU=0.7 | | |
|---|---|---|---|
| | near | mid | far |
| MonoDTR [8] | 34.52 | 6.73 | 0.19 |
| MonoCon [7] | 35.19 | 6.74 | 0.40 |
| MonoDETR [9] | 32.62 | 7.20 | 0.40 |
| MonoHBD (Ours) | **36.02** | **7.70** | **0.52** |

Table 4: The depth prediction performance on the KITTI validation set.

| Method | $\delta < 1.25 \uparrow$ | Abs Rel $\downarrow$ | Sq Rel $\downarrow$ | RMSE $\downarrow$ | RMSE$_{log}$ $\downarrow$ |
|---|---|---|---|---|---|
| MonoDETR [9] | 0.2184 | 1.0210 | 16.6006 | 13.0484 | 0.7395 |
| baseline | 0.2575 | 0.6952 | 10.5509 | 11.1477 | 0.5699 |
| MonoHBD (Ours) | **0.8680** | **0.1422** | **1.1859** | **3.4897** | **0.3515** |

First, we observed a significant improvement when incorporating the calibration matrix $K$ into the depth network. Next, we introduce the HDM structure and further boost the performance, indicating the effectiveness of depth offsets. Adopting RFL for depth map regression also contributes to the performance gain. Lastly, leveraging the BDE, our model achieves a remarkable improvement compared to the baseline, showcasing the effectiveness of each proposed module.

**Comparison of the performance at different depth ranges.** We evaluate model performance at three depth ranges: "near" (0m-20m), "mid" (20m-40m), and "far" (40m-60m). Table 3 shows the performance of different models at various depth ranges on the KITTI validation dataset. Our proposed method surpasses the second-best model by a large margin (0.83 (+2.36%), 0.5 (+6.94%), 0.12 (+30.00%)) across all depth ranges. These results demonstrate the efficacy of our approach in accurately detecting objects, especially those located at greater depths.

**Comparison of the depth prediction performance.** Following the depth prediction metric from [35], we evaluate the threshold accuracy ($\delta_i$), absolute relative error (Abs Rel), squared relative error (Sq Rel), root-mean-square error (RMSE), and root-mean-square log error (RMSE$_{log}$) on the KITTI validation set. To ensure a fair comparison, we only compare our results to other works utilizing Foreground Depth Map [9]. The results are presented in Table 4. Considering the performance of [9] is low, we additionally include our baseline model for comparison. Our model significantly outperforms other models, substantially reducing prediction errors.

**Limitations.** As presented in Table 3, the performance decreases rapidly as the depth increases. Besides, our method hardly detects occluded objects due to the lack of available information. To overcome these limitations, we suggest future works explore the use of temporal information to improve scene-level understanding and enhance the detection of occluded objects.

## 5   Conclusion

In this paper, we explore the limitations of existing depth supervision methods and propose a novel framework specifically designed for monocular 3D object detection. Our proposed modules, including the HDM structure and the BDE module, effectively capture depth relationships and enhance localization accuracy. We also introduce a specialized depth map loss to mitigate the issue of imbalanced loss weighting. Through extensive experiments on the KITTI dataset, our approach has demonstrated superior performance compared to state-of-the-art methods while achieving real-time detection capabilities. Our work contributes to advancing the monocular 3D object detection field by providing a new comprehensive perspective of depth-guided methods.

**Acknowledgments**

This work was supported in part by National Science and Technology Council, Taiwan, under Grant NSTC 111- 2634-F-002-022, and Mobile Drive Technology Co., Ltd (MobileDrive). We are grateful to the National Center for High-performance Computing.

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

Table 5: The detection performance of Pedestrian and Cyclist categories on the KITTI test set. The best results are highlighted in red, while the second best ones are highlighted in blue.

| Method | AP 3D Ped.@IoU=0.5 | | | AP 3D Cyc.@IoU=0.5 | | |
|---|---|---|---|---|---|---|
| | easy | mod | hard | easy | mod | hard |
| MonoDLE [31] | 9.64 | 6.55 | 5.44 | 4.59 | 2.66 | 2.45 |
| CaDDN [10] | 12.87 | 8.14 | 6.76 | **7.00** | **3.41** | **3.30** |
| MonoFlex [33] | 9.43 | 6.31 | 5.26 | 4.17 | 2.35 | 2.04 |
| GUPNet [3] | **14.95** | **9.76** | **8.41** | 5.58 | 3.21 | 2.66 |
| MonoDTR [8] | **15.33** | **10.18** | **8.61** | 5.05 | 3.27 | **3.19** |
| MonoCon [7] | 13.10 | 8.41 | 6.94 | 2.80 | 1.92 | 1.55 |
| MonoHBD (Ours) | 12.40 | 8.16 | 6.80 | **5.75** | **3.42** | 3.08 |

## A    Details of Depth-aware Transformer

Previous works have proposed various depth-aware transformers [36, 27, 9, 8] and achieved great results. In our work, we follow the settings from [9], which contains a visual encoder, a depth encoder, and an object decoder that merges cross-view attention and cross-depth attention to generate depth-aware features. Inspired by [27], we first feed the learnable object queries $F_Q \in \mathbb{R}^{N \times C}$ to the transformer, where $N$ is the number of underlying 3D bounding boxes. In each transformer layer, the queries interact with image features and depth embeddings through the cross-view attention followed by the cross-depth attention. After $L$ layers, we obtain the depth-aware object queries.

## B    Implementation Details

We utilize the AdamW optimizer with a weight decay of $10^{-4}$ and an initial learning rate of $10^{-5}$. To warm up the training process, we employ cosine warm-up scheduling for the first 5 epochs. Afterward, the learning rate is set to $2 \times 10^{-4}$ and decays at epoch 125 with a rate of 0.1. We set $[d_{min}, d_{max}] = [10^{-3}, 60]$ and the number of depth bins $N_D = 80$ for the LID method. The input image size is $1280 \times 384$ for both training and testing. Following [3], we apply the photometric distortion, random horizontal flipping, and random cropping techniques. The random scaling factor is 0.4 while the random cropping factor is 0.25. Our model predicts up to $N = 50$ bounding boxes for each image sample without requiring Non-Maximum Suppression (NMS).

## C    Additional Results

**Performance of the Pedestrian and Cyclist categories.** We outline the performance of our model on the Pedestrian and Cyclist categories of the KITTI test set in Table 5. In the Cyclist category, our model achieves the highest AP 3D score at the "moderate" level and the second highest at the "easy" level. For the Pedestrian category, our model exhibits competitive performance, closely rivaling leading models such as GUPNet [3] and MonoDTR [8]. We argue that the superior performance of the Cyclist category can be attributed to the HDM structure. However, the performance in the Pedestrian category is slightly lower, possibly due to the less significant depth offsets in pedestrians compared to cars and cyclists.

**Comparison of different types of ground truth depth maps.**    Table 6 shows the performance of the KITTI validation set of utilizing different types of ground truth depth maps. Previous works, such as CaDDN [10], generate dense depth maps by projecting point clouds onto the image plane followed by depth completion [37]. On the other hand, Foreground Depth Map [9] is built by the ground truth 2D bounding boxes and object center depths. Our model performs better with the Foreground Depth Map. We argue that the traditional dense depth map only contains the depth of the object surface, which cannot be directly employed in 3D object detection tasks. In contrast, Foreground Depth Map stores the depth of the object center, making depth features generate more relevant information for the detection head.

Table 6: Comparison of utilizing different ground truth depth maps on the KITTI validation set.

| Depth Map Type | AP 3D Car@IoU=0.7 | | |
|---|---|---|---|
| | easy | mod | hard |
| Dense Depth Map [10, 8] | 21.37 | 16.08 | 13.14 |
| Foreground Depth Map [9, 24] | **23.34** | **16.78** | **13.70** |

**Qualitative Results.** Figure 4 illustrates the qualitative results of the KITTI validation set. Our model demonstrates accurate object localization for objects at near and moderate distances. However, it faces challenges when dealing with heavily occluded objects or those located far from the image center. These challenges arise due to missing information on the image and object distortion, which are common limitations of monocular-based methods.

To further demonstrate the effectiveness of our model, particularly in detecting distant objects, we compare our prediction results with those of MonoDTR [8]. Figure 5 showcases the 3D detection results of our model and MonoDTR in both the image and LiDAR coordinates. Our model exhibits superior performance in localizing distant objects compared to MonoDTR. Moreover, we are able to detect far objects that MonoDTR fails to detect.

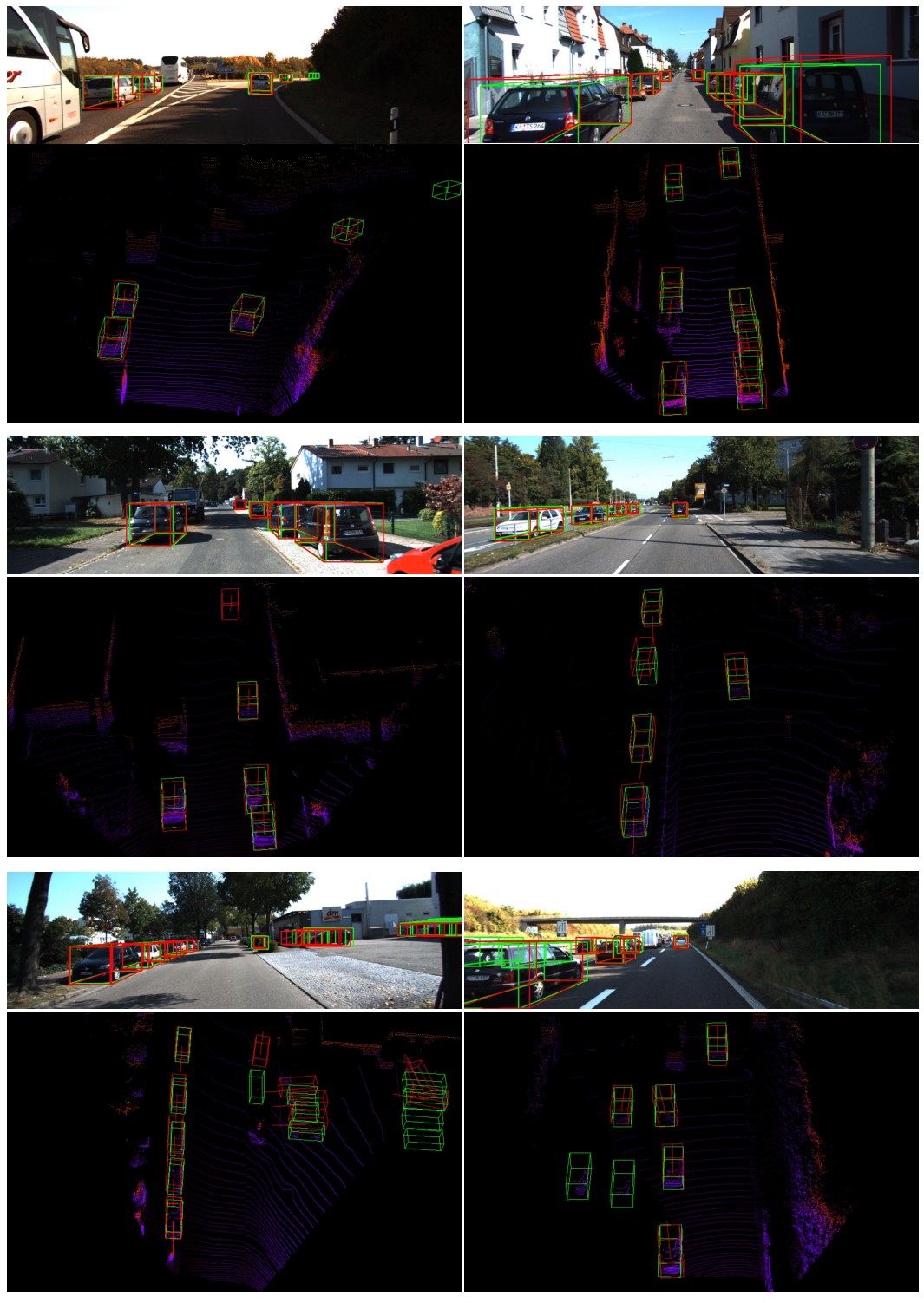

Figure 4: Qualitative results on the KITTI validation set. The red bounding boxes represent the prediction of our model, while the green ones denote the ground truth bounding boxes. Best viewed in color and zoom-in.

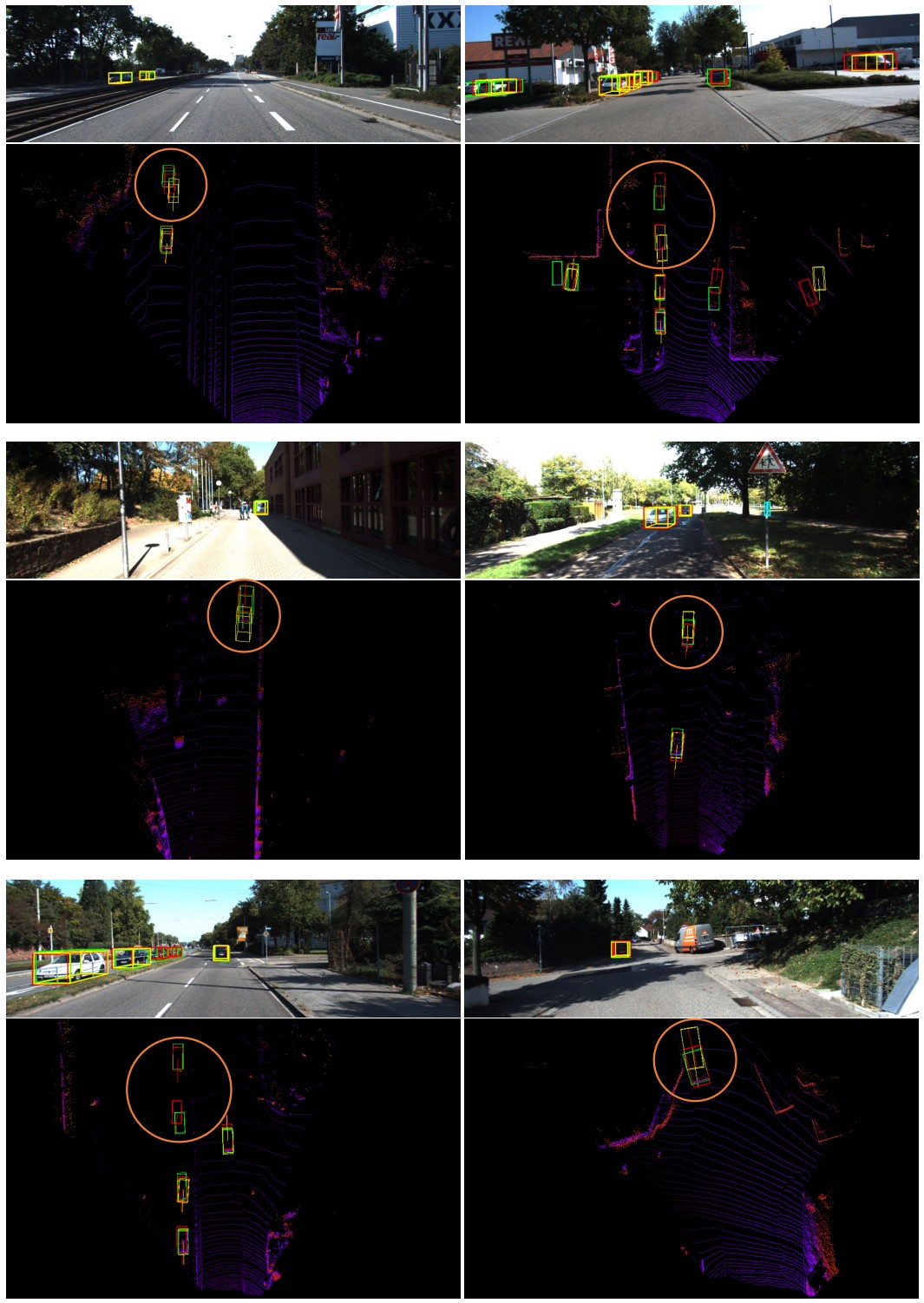

Figure 5: Comparison of our MonoHBD and MonoDTR [8] on the KITTI validation set. The green bounding boxes represent the ground truth boxes. The red, yellow boxes denote the prediction of our model and MonoDTR, respectively. Best viewed in color and zoom-in.

