# OpenReview forum: "Revisiting Depth-guided Methods for Monocular 3D Object Detection by Hierarchical Balanced Depth"
_robot-learning.org/CoRL/2023/Conference — CoRL 2023 Poster_

### Official Review · Reviewer_BqXK · 2023-07-11

**Confidence:** 5
**Originality:** Good
**Technical Quality:** Good
**Clarity Of Presentation:** Good
**Impact:** 3

**Recommendation:**

Weak Accept: I recommend accepting the paper, but will not argue for my recommendation if the majority of other reviewers have a different opinion.

**Review:**

In terms of pixel-level depth estimation, the authors’ analysis is reasonable. Imbalanced loss weighting among objects with different depths affects the performance for distance objects. The proposed Balanced Depth Extractor is a way to tackle this problem. The first branch predicts the pixel-wise depth distribution across the entire scene, maintaining geometric relationships among objects. The second branch concentrates on the depth of foreground objects and ensures that the object size difference is removed, leading to more balanced loss weighting for each object regardless of their depth.

The experiment also validated the contribution of each module. Table 4 is quite important since it clearly shows the advantage of this method for distant objects.


**Quality Of The Limitations Section:**

Limitations are addressed clearly

**Questions For Rebuttal:**

- Why camera parameters bring the most performance gain.
- How do you do data augmentation (scaling, rotation) and how are the camera parameters changed?
- The IoU is influenced by many factors. If the authors mainly claimed their contribution to depth estimation, there lacks some experiments to isolate the depth from other factors and only evaluate the depth accuracy.


**Robotics Focus:**

Highly relevant to robotics but no hardware experiments

**Summary Of Paper:**

The paper presents a monocular 3D object detection method. The authors identify inaccurate depth estimation as the primary issue, caused by the imbalance of loss weighting between near and distant objects, in traditional pixel-wise depth map loss. They introduce a Hierarchical Depth Map (HDM) structure that improves object localization accuracy through depth bins and depth offsets. They also proposed the Balanced Depth Extractor (BDE) module that captures both scene-level and object-specific depth details, taking into account geometric characteristics through the incorporation of camera calibration parameters. The model shows impressive results on the KITTI 3D object detection benchmark.

**Summary Of Recommendation:**

A valid paper with some useful insights.

---

### Official Review · Reviewer_yA3D · 2023-07-20

**Confidence:** 5
**Originality:** Very Good
**Technical Quality:** Good
**Clarity Of Presentation:** Good
**Impact:** 4

**Recommendation:**

Weak Accept: I recommend accepting the paper, but will not argue for my recommendation if the majority of other reviewers have a different opinion.

**Review:**

# Strengths:

* The proposed RoIAlign module is interesting and novel, leverages the idea from Mask-RCNN to address the balancing problem in depth prediction.

# Weaknesses:

* The idea of the HDM looks trivial and there are lots of existing works shares a similar idea.

* The detection performance on KITTI dataset looks not too impressive

**Quality Of The Limitations Section:**

Additional details required

**Questions For Rebuttal:**

1, Could you share more insights behind the weighting factor design of the Regression Focal Loss?

2, Have you evaluated the depth prediction accuracy on the KITTI dataset?

3, there are some other related depth-guided monocular 3D detection works could be discussed in this paper:

[1] Depth Estimation Matters Most: Improving Per-Object Depth Estimation for Monocular 3D Detection and Tracking

[2] "Advancing self-supervised monocular depth learning with sparse liDAR." Conference on Robot Learning. PMLR, 2022.

[3] End-to-End Pseudo-LiDAR for Image-Based 3D Object Detection

**Robotics Focus:**

Highly relevant to robotics but no hardware experiments

**Summary Of Paper:**

This paper proposed an end-to-end monocular depth estimation and 3D object detection framework. The key contributions are as follows:

1, This paper identifies a fundamental balancing problem of the conventional depth loss formulation, and proposed a novel ROIAlign based method to address this issue.

2, This paper proposed a regression focal loss (RFL) and a hierarchical depth map (HDM) for better depth estimation.

**Summary Of Recommendation:**

This paper identifies several unique and important problems in the depth prediction task, and accordingly proposed methods to solve them. I suggest a weak accept rating for this paper.

---

### Official Review · Reviewer_shF4 · 2023-07-25

**Confidence:** 5
**Originality:** Very Good
**Technical Quality:** Very Good
**Clarity Of Presentation:** Excellent
**Impact:** 4

**Recommendation:**

Weak Accept: I recommend accepting the paper, but will not argue for my recommendation if the majority of other reviewers have a different opinion.

**Review:**

Overall, this paper is very well written, easy to follow, and the proposed method technically sound. The authors did a good job of articulating the problem statement with a good visualization in Figure 2. The ablation result clearly shows improvements after adding each of the proposed modules.

Please address the following for the rebuttal,

1. Currently, the model is trained and tested on the KITTI-3D dataset only. Would it be possible to test on other datasets for generalization and report the numbers?

2. Since the camera calibration matrix is being used as an input, I am keen to know what will happen to the results when tested with different resolutions/crops of the same image or a completely different camera setup.

3. As the majority of the paper is about improving the depth-estimation performance, I was wondering if the authors did any evaluation on the estimated depth directly.


**Quality Of The Limitations Section:**

Limitations are addressed clearly

**Questions For Rebuttal:**

Please see above for the questions.

**Robotics Focus:**

Highly relevant to robotics but no hardware experiments

**Summary Of Paper:**

This paper proposes a monocular 3D object detection method that benefits from improved depth estimation. The authors argue that the commonly used pixel-wise depth map loss is imbalanced for distant and near objects. To mitigate this, they extended the conventional discretization method by estimating depth offsets. Also, they further proposed to add Regression based Focal Loss (RFL) to account for the imbalance. By doing so, they were able to improve the depth estimation thereby improving the monocular 3D object detection metrics, AP BEV and AP 3D.

**Summary Of Recommendation:**

This paper did a good job at addressing the imbalance problem in depth estimation that directly shows the impact on 3D object detection. Adding camera calibration matrix embeddings to the depth features is a good idea. Lastly, estimating the depth offsets and mitigating the problem of the wider depth-bin problem is also good. However, I would like to know more about the generalization aspect of the method from the rebuttal.

---

### Decision · Program_Chairs · 2023-08-30

**Decision:**

Accept (Poster)

**Comment:**

The paper identifies a fundamental balancing issues with the conventional depth loss formulation, and based on that proposes a novel depth estimation and 3D object detection method. The methods outperforms previous 3D object detection methods on the KITTI data set.

The reviewers generally appreciate the technical contributions. There were only minor questions raised that the authors addressed in their
rebuttal.